# Profitability of Chemically Cross-Linked Collagen Scaffold Production Using Bovine Pericardium: Revaluing Waste from the Meat Industry for Biomedical Applications

**DOI:** 10.3390/polym15132797

**Published:** 2023-06-23

**Authors:** José Arturo de la Cruz Bosques, José de Jesús Ibarra Sánchez, Birzabith Mendoza-Novelo, Juan Gabriel Segovia-Hernandez, Carlos Eduardo Molina-Guerrero

**Affiliations:** 1Departamento de Ingenierías Química, Electrónica y Biomédica, Universidad de Guanajuato, Lomas del Bosque 103, Col. Lomas del Campestre, León 37150, Guanajuato, Mexico; delacruzlj2015@licifug.ugto.mx (J.A.d.l.C.B.); chuy_lindo3@hotmail.com (J.d.J.I.S.); 2Departamento de Ingeniería Química, Universidad de Guanajuato, Noria Alta s/n, Col. Noria Alta, Guanajuato 36000, Guanajuato, Mexico; g_segovia@hotmail.com

**Keywords:** animal biomass, circular bioeconomy, process development, sustainability

## Abstract

The meat industry generates a large amount of waste that can be used to create useful products such as bio-implants, which are usually expensive. In this report, we present an economic analysis of a continuous process for large-scale chemically cross-linked collagen scaffold (CCLCS) production in a Mexican context. For this purpose, three production capacities were simulated using SuperPro Designer^®^ v 12.0: 5, 15, and 25 × 10^3^ bovine pericardium units (BPU) per month as process feedstock. Data indicated that these capacities produced 2.5, 7.5, and 12.5 kg of biomesh per batch (per day), respectively. In addition, Net Unit Production Costs (NUPC) of 784.57, 458.94, and 388.26 $USD.kg^−1^ were obtained, correspondingly, with selling prices of 0.16 ± 0.078 USD.cm^−2^, 0.086 ± 0.043 USD.cm^−2^, and 0.069 ± 0.035 USD.cm^−2^, in the same order. We found that these selling prices were significantly lower than those in the current market in Mexico. Finally, distribution of costs associated with the process followed the order: raw materials > facility-dependent > labor > royalties > quality analysis/quality control (QA/QC) > utilities. The present study showed the feasibility of producing low-cost and highly profitable CCLCS with a relatively small investment. As a result, the circular bioeconomy may be stimulated.

## 1. Introduction

The meat industry generates a large amount of waste, which can be used to create useful products such as bio-implants that are usually expensive. Some of these materials are used in wound healing, dental repairment, bone repairment, skin, vessels, cartilage, nervous system, cardiac, liver, and injectables, among others [1,2,3,4]. Unfortunately, the cost of the production of soft tissue scaffolds is usually high as it involves the expenses of research and development, product safety, clinical trials, storage, and marketing, among others [5]. Currently, different companies, including Skin & Healhcare, Kowa Co., Terumo Corp., Tei Bioscience Inc., and Groupe Perouse Plastie, among others, produce biomeshes using bovine collagen. These materials are used as soft tissue replacement in the treatment of chronic ulcers or thickness wounds, among others, with costs that vary from USD/cm^−2^ 4 to 15 [5]. Unfortunately, developing countries usually pay higher prices because of additional fees related to transportation and import tariffs. For this reason, the most vulnerable people do not have access to these types of treatments. In order to reduce the total production cost (TPC), a potential solution is large-scale manufacturing. However, several considerations must be taken into account.

An alternative to producing these biomaterials, i.e., biomeshes, is using bovine pericardium (BP), which is rich in type I collagen. In Mexico, BP represents waste from slaughterhouses. BP tissue displays high flexibility, mechanical resistance, and the appropriate thickness that makes it a unique material for the production of devices that can be inserted into the human body due to its ability to contract and expand [5,6]. Several authors have reported the use of BP biomeshes in heart valvules and decellularized scaffolds for the regulation of blood sugar level in patients with Type I diabetes [7,8].

According to the National Institute of Statistics and Geography (INEGI), cattle slaughtered in Mexico totaled 156,936 in February 2022 [9]. Previously, in 2014, a total of 2,052,303 cattle slaughtered was reported. This same year, the five states with the highest cattle slaughter (bovine.yr^−1^) were Jalisco (332,149), Michoacan (211,327), Guanajuato (174,711), Mexico (119,877), and Veracruz (118,069). In addition, the smallest numbers (bovine.yr^−1^) corresponded to Tlaxcala (8402), Quintana Roo (9924), Baja California Norte (10,495), Nuevo León (11,427), and Baja California Sur (12,910) [9]. Thus, slaughterhouses represent a potential source of type I collagen. In Mexico, some new policies promote the transition to a circular economy, which will allow the use of slaughterhouse waste for different purposes including the production of biomaterials [10].

In this context, process simulation can be helpful in evaluating the profitability of the techniques used to transform type I collagen into biomeshes. Process simulation is used in equipment design and to calculate energy costs as well as the profitability of processes at different scales. Recently, process simulation has been used to evaluate several processes in order to determine the optimal operational conditions that allow profitability and reduce the need for government subsidies [11,12,13,14,15].

The aim of the present investigation was to develop a large-scale process for biomesh production in a Mexican context. For this purpose, we selected three scenarios where 5000, 15,000 and 25,000 BPU.mth^−1^ were used as raw materials. The proposed process includes decellularization, hydrolysis, crosslinker synthesis, and crosslinking, with a focus on volume of production (USD.kg^−1^). In order to compare the results with data provided in commercial products, volume of production was converted to USD.cm^−2^. Furthermore, a sensitive analysis to determine the minimal biomesh selling price was carried out and results were compared with actual selling prices. Finally, we proposed alternatives to improve the economic performance of biomesh production using type I collagen from bovine pericardium from an engineering perspective.

## 2. Materials and Methods

### 2.1. Bovine Pericardium Composition

The pericardium is composed of simple squamous epithelium and connective tissue. It is rich in type I collagen as well as glycoproteins and glycosaminoglycans (GAG) in addition to constituent cells. Collagen is arranged at different levels ranging from fibrils to laminae, fibers, and fiber bundles [16]. The composition of the pericardium used in this research was obtained from [17]. Briefly, the composition on a dry basis is (g.kg^−1^): protein (40), collagen (770), elastin (40); the rest are sugars such as galactosamine (4.8), glucosamine (7), and glucuronic acid (1.6). Finally, the weight for each unit of bovine pericardium was assumed to be 15 g.

### 2.2. Process Description

The process was simulated in steady state using the SuperPro Designer v.12.0^®^ (SPD) process simulation software considering 330 days of operation. The process was divided into four stages: (i) conditioning (or tissue decellularization), (ii) crosslinker synthesis (formation of PEG-1000 oligourethanes and hexamethylene disocyanate), (iii) crosslinking (crosslinking reaction between the decellularized tissue and the crosslinker), and (iv) post-treatment (washing, drying, and packaging). The bench scale experiments were performed using five agitation tanks, four continuous stirred reactors (CSTR), two centrifuges, and one freeze dryer (See Appendix B, Figure A4, Table A1). The volumes of the 316 stainless steel tanks were calculated using mass balance. Additionally, the Guthrie method, adapted in SPD, was used to determine tank cost. Tank parameters are reported in Appendix B and Figure 1 shows the block flow diagram of the process used in the present investigation.

#### 2.2.1. Decellularization Stage

Bovine pericardium units (BPU) were decellularized following the procedure reported in [18]. The process flow diagram for biomesh production is presented in Figure A4 (Appendix B). In the first stage (decellularization), bovine pericardium (animal tissue) was soaked with a phosphate buffered saline solution (PBS) and Triton-X-100 (a non-ionic surfactant useful in protein dissolution) (NCBI, 2021). In order achieve repulsive forces and, in turn, tissue swelling, BPU were treated with an alkaline solution [18]. Later, samples were placed in sterile PBS at a ratio of 7 mL solution per g of tissue (PBS, 30 mM, pH 7.4, 0.9% NaCl) containing Triton-X (1% *v*/*v*) and allowed to stay at room temperature for 48 h. The surfactant was replaced after 24 h with a fresh solution. The mixture was allowed to stay another 24 h and, after this period, the samples were washed with PBS (P3) and subsequently dried using a centrifuge (current S-102 towards P1). Then, the solution was transferred to a stirring tank (stream S-101 towards P19), where DNases and RNases (Nucleases) were added to remove residual nucleic acids (DNA and RNA) from damaged cells. At this stage, the tissue was placed in a Tris HCl 10 mM solution at pH 7.6 (2.5 mM MgCl_2_, 0.5 mM CaCl_2_) containing 0.2 mg.ml^−1^ DNase and 0.02 mg.ml^−1^ RNase. Then, samples were transferred to an agitation tank (current S-104 towards P20) and washed with a PBS/NaCl solution. To get rid of chemical residues and cell debris, tissues were washed for 24 h at room temperature under continuous stirring. Samples were further dried using a centrifuge (stream S-105 towards P4) (Figure A4).

#### 2.2.2. Crosslinker Synthesis

The crosslinker used in the present experiments was synthesized through an oligomerization reaction between PEG-1000 and hexamethylene diisocyanate (HDI) [18,19] inside the R-101 reactor (Figure A4). Briefly, the molten PEG was reacted with the diisocyanate (HDI) at an NCO:OH molar ratio of 4.0:1.0 for 2 h at 100 °C. This ratio was chosen to obtain end isocyanate groups in products and to ensure the subsequent addition reaction with collagen. Although the main oligomer structure corresponds to a blocked trimer structure, i.e., isocyanate–PEG–isocyanate (I–PEG–I) chains, as shown in Appendix A (Figure A1 and Figure A2), chain extension reactions are promoted [20]. Thus, corresponding equations of the reactions carried out in R-101 and R-102 equipment are shown in Table 1.

After oligomerization, the temperature was reduced to 60 °C. The energy used for this purpose was calculated using a heat exchanger (current S-110 to P6) to a second reactor (current S-111 to R-102). To avoid further oligourethane polymerization, a saturated sodium bisulfite (40% NaHSO_3_) solution was added, and temperature was maintained at 40 °C for 2 h (Figure A4. Oligourethane blocking) (Table 1) [18].

The oligomerization reaction produced trimers, pentamers, and heptamers (see Table 1). However, trimers were more abundant than the rest of the polymers. For this reason, they were selected for biomesh production [19]. After oligomer blockage, the mixture was diluted with water to achieve a concentration of 40% (solid weight) (current S-113 towards S-115). Later, silica particles were added to the oligourethane using a tetraethyl orthosilicate-ethanol solution (current S-103 towards P7) (Silica SiO_2_ [TEOS]), and the mixture was stirred for 18 h at 25 °C (room temperature, RT) to achieve complete dispersion of colloidal silica in the prepolymer solution. Finally, the crosslinker was transferred to a stirring tank (current S-120 towards P13) and peroxide (H_2_O_2_, 30% weight at RT) was added to lower the pH. Oligomerization and blocking steps are shown in Figure A4, while the addition of silica is represented in [21].

#### 2.2.3. Crosslinking

After decellularization and crosslinker synthesis, a crosslinking reaction was performed. For this purpose, the S-107 and S-108 streams were mixed in the R-103 reactor. Additionally, magnesium oxide (1% *v*/*v*) was added until pH 8.2 to “unblock” the oligourethane and allow the crosslinking reaction to proceed. In this way, the tissue and the crosslinking agent reacted, producing a collagen network. The oligourethane (20% weight with respect to the tissue) was stirred for 3 h at RT. Magnesium oxide and the crosslinking reaction were prolonged for 10 h under stirring conditions at RT. The crossover methodology was followed as indicated in [18,19]. Corresponding reactions are shown in Table 1 and Figure A1 and Figure A2.

#### 2.2.4. Washing

The cross-linked tissue (R-103 reactor) was transferred to the agitation tank (P16) through the S-106 stream. Later, the reaction mixture was treated with 0.03M EDTA at 4 °C to remove any residual agent.

#### 2.2.5. Freezing and Drying

Once washed with EDTA (0.03 M at 4 °C), the crosslinked tissue was moved to a freeze dryer (stream S-117 from P16 to P17) where it was frozen-dried at −70 °C to preserve its properties. After this process, the dry product was ready for packaging (current S-119) (See Figure A3).

### 2.3. Scenarios

In this work, UPB of 5000, 15,000, and 25,000 BPU.mth^−1^ were selected considering the amount of BPU produced in slaughterhouses in Mexico [9].

### 2.4. Financial Investment and Assumptions

#### 2.4.1. Net Present Value Method (NPV)

The biomesh Net Unit Production Cost (NUPC) per kilogram was calculated using the dynamic flow of capital analysis (DFCA) which uses the Net Present Value (NPV) = 0 (e.g., i is equal to the internal rate of return (IRR)). This methodology is used by SPD and explained in [16]. Briefly, the NPV is a function of the selected capacity and feed intervals for a fixed financing and production condition. The year of 2021 was selected for the analysis.
(12)$$\mathrm{NPV}=\frac{Cash\;flow\;\left({\left(1+i\right)}^{n}-1\right)}{i{\left(1+i\right)}^{n}}+\frac{working\;capital}{{\left(1+i\right)}^{n}}-investment\;$$
(13)Cash flow=Cash inflow−Cash outflow
(14)Cash outflow=Direct production cost+taxes+loan annuity;
(15)$$Direct\;costs\;\left(DC\right)=f_{1}\left(equipment\;cost\right);$$
(16)$$Indirect\;costs\;\left(IC\right)=f_{2}\left(equipment\;cost\right);$$
(17)Fixed Capital Investment (FCI)=DC+IC;
(18)$$star\;up=f_{3}\left(FCI\right);$$
(19)$$working\;capital\;\left(WC\right)=f_{4}\left(FCI+start\;up\right);$$
(20)total capital investment (TCI)=FCI+WC+start up;
(21)investment=FCI−loan.

Using the NPV, a sensitivity analysis was performed to find the minimal selling price in the three different scenarios.

#### 2.4.2. Economic Considerations

Financial investment and assumptions were considered according to SuperPro Designer—User’s Guide. The process considered an operating time of 330 days/year. In addition, it was assumed that 30% of the capital was borrowed for 10 years at an interest rate of 6% and an inflation factor of 4%. Patent royalties of 4% and useful life and project construction of 15 and 1 years, respectively, were selected. In addition, a linear depreciation of the equipment was considered. Federal taxes were set up at a 40% gain. Additionally, a cost of USD 0.1 kWh for the electricity used in heating, cooling, agitation, and centrifugation was established, according to the Central Region Office of the Federal Electricity Commission (CFE) of Mexico [22]. With respect to human resources, an hourly wage of USD 6.25 in 8-h schedules with no rotating shifts was selected. In addition, 1 month included 30 days and 2% TPC was considered as the cost of biomesh sterilization. The cost of the reagents used in biomesh production are shown in Appendix C (Table A3).

## 3. Results

An economic analysis for the production of cross-linked collagen from bovine pericardium was carried out. For this purpose, production capacities of 5, 15, and 25 × 10^3^ BPU.mth^−1^ were selected. Figure 2 presents the profitability analysis for a production capacity of 5 × 10^3^ BPU.mth^−1^. Similar analyses were performed for production capacities of 15 and 25 × 10^3^ BPU.mth^−1^. These data are presented in Appendix C. Figure 2A depicts the increase in process capacity (annual throughput) against NUPC. As expected, the cost of production showed the typical exponential decay of scaling up. According to the results, an NUPC of USD 12,836.58 per kg biomesh was obtained at a process capacity of 30 kg. yr^−1^. In addition, a process capacity of 750 kg.yr^−1^ (2.5 kg.day^−1^) resulted in an NUPC value of USD 784.57 per kg biomesh. Figure 2B displays the Facility Dependent Product Cost (USD.kg^−1^) against process capacity (annual throughput). Similarly, an exponential decay behavior was observed with values of USD 7327.36 and USD 318.58 for capacities of 30 and 750 kg.yr^−1^, respectively. These results confirm that an increase in capacity decreased the NUPC and facility-dependent cost. Data on revenues and NP against annual throughput are presented in Figure 2C. Herein, a positive NP resulted when the process capacity exceeded 360 kg per year. However, the NP should be confirmed with a positive IRR value. Indeed, positive IRR values of 1.55% and 0.58% before and after taxes, respectively, were determined when process capacity was 510 kg.yr^−1^ (Figure 2D). The IRR value is a metric used in financial analysis to estimate the profitability of potential investments. The IRR value is a discount rate that makes the net present value (NPV) of all the cash flows equal to zero in a discounted cash flow analysis. Our data also indicated that the maximum process capacity (750 kg.yr^−1^ or 2.5 kg.day^−1^) provided IRR values of 16.66% and 13.09% before and after taxes, correspondingly. In addition, an ROI of 20.41% was obtained for a process of 750 kg.yr^−1^. This value confirms that the process proposed in the present investigation is profitable.

Table 2 presents data on total capital investment (TCI), operating cost (OC), revenues, NUPC, ROI, IRR, payback time, and NPV for the production of biomesh in three different scenarios. As expected, the investment costs followed the order 5 < 15 < 25 × 10^3^ BPU.mth^−1^. According to these results, production of 2.5, 7.5, and 12.5 kg of biomesh per day can be potentially achieved. Thus, when production capacity increased from 5 × 10^3^ BPU.mth^−1^ to 15 × 10^3^ BPU.mth^−1^, TCI augmented 28.21%. However, the TCI of a production capacity of 25 × 10^3^ BPU.mth^−1^ was only 16.05% higher than that of 15 × 10^3^ BPU.mth^−1^. It was also found that NUPC decreased as production capacity increased, with NUPC values of 784.57, 458.94, and 388.26 USD.kg^−1^ for biomesh production capacities of 5, 15, and 25 × 10^3^ BPU.mth^−1^, respectively.

In the present work, an ROI value of ≈20% was selected. The ROI is a performance measure used to evaluate the efficiency or profitability of a given investment and, in general, an acceptable value is over 5%. Our data indicated a payback time of about 5 years, a period that can be shortened whenever a higher price is considered. In addition, a higher selling price can reduce the payback time, affecting the ROI and the profitability of the process. The results in Table 2 indicate that, in the three scenarios analyzed in our work, IRR values of about 13% were obtained. This means that, in all the cases, profitability is possible. Finally, in all the cases, positive NPVs were obtained. In addition, it was observed that, as NPV increased, process capacity improved. NPV corresponds to the present value of the cash flow at the required ROI compared to the initial investment. Thus, the initial TCI will be depreciated by ≈40% after ≈5 years in the three scenarios.

In order to obtain the minimum selling price per kg of biomesh, a sensitivity analysis was performed. Figure 3 presents the NPV and biomesh selling price in different scenarios. Figure 3A shows the results for a production capacity of 5 × 10^3^ BPU.mth^−1^ with selling prices varying between USD 350 and 500/kg. Positive unit product revenues of USD 1043.75, 579.86 and 463.89.kg^−1^ corresponded to selling prices of USD 400, 450, and 500.kg^−1^, respectively. On the other hand, a selling price of USD 350.kg^−1^ resulted in a negative NPV. Thus, profitability is possible with prices above USD 400.kg^−1^. Process capacities of 15 and 25 × 10^3^ BPU.yr^−1^ yield minimum selling prices of USD 250 and 200.kg^−1^, correspondingly. Below these values, the processes will be considered unprofitable.

In Figure 4, costs distributions are presented (Please, see Table A2 for fixed capital estimate summary). Data showed that the cost of raw materials increased as production size increased. In addition, the fraction corresponding to the cost of labor decreased as process capacity augmented. In this particular case, the size of the plant and the intended production do not require additional workers. We proved that facility-related costs are higher at relatively small production capacities. This is associated with the fact that the use of facilities is similar in the three proposed scenarios; however, for the use of facilities for the smallest process, it implies the same effort as for the other two scales assayed. The costs related to quality assurance (QA) and quality control (QC) are relatively elevated at small scales, as observed in the scenario with the lowest production capacity. The QA/QC costs were calculated as a 1.4, 1.97 and 3.45 percentage of the total labor cost (TLC) of the 5, 15, and 25 × 10^3^ BPU.mth^−1^ capacities, respectively. In addition, utilities expenses represent a small percentage of the operating costs. This item is mainly associated with the costs for heating, cooling, and centrifugation, which do not represent a high energy consumption during the biomesh manufacturing process.

### Comparison with Actual Commercial Products

Currently, different companies also offer collagen-based products of bovine or porcine origin for the treatment of burn injuries. Those of bovine origin cost between USD 4 and 8 per cm^2^, while porcine-based products sell at prices between USD 3 and 14 per cm^2^ [5]. These biomeshes display an area of 1 cm^2^ and a thickness of 0.5 mm ± 0.25 mm [5,23]. In addition, 1 cm^2^ units present density, weight, and volume of 3 g.cm^−3^, 0.15 ± 0.075 g, and 0.05 ± 0.025 cm^3^. These data were used to compare the prices of 1 g of product in the market with those obtained in this research. Our results indicated that a production capacity of 5 × 10^3^ BPU.mth^−1^ yield a unit product revenue of USD 1043.75.kg^−1^ (USD 0.16 ± 0.078.cm^−2^). In addition, production capacities of 15 and 25 × 10^3^ BPU.mth^−1^ represent revenues of USD 0.086 ± 0.043.cm^−2^ and USD 0.069 ± 0.035.cm^−2^, respectively. The minimum selling price reported by Kowa Co was USD 4.cm^−2^ [5]. Herein, we demonstrated that the continuous production of collagen from BP proposed by our research group drastically reduced the TPC and selling price of collagen-based biomesh. In consequence, low-income consumers will eventually have more access to these types of products.

## 4. Discussion

The results obtained in our experiments indicated that the production of biomeshes using the proposed process is economically feasible. Moreover, the commercialization of decellularized tissues for additional purposes is also recommended. From the point of view of process engineering, the process is feasible and highly profitable. For this reason, it can be an attractive investment opportunity for those interested in the development of new biomaterials. Difficulties during the collection of bovine pericardium samples may be a downside of the process as they may cause delays in biomesh production. Recently, the Mexican Senate announced an initiative to promote the establishment of circular economy practices. This announcement may encourage slaughterhouses in this country to generate additional economic benefits from bovine pericardium, which is currently considered a waste. A strong involvement of the social and academic sectors will be required for the development of biomaterials from municipal waste.

## 5. Conclusions

A new methodology to manufacture biomesh from BP through a decellularization and crosslinking process using oligourethanes was proposed herein. The process is economically profitable at commercial scales. Data indicated that the three scales analyzed in the present study would require a relatively small amount of investment. This number varies between USD 1 and 1.7 M with a very acceptable rate of return (ROI) of around 20% and investment return times of no more than 5 years. One of the most important conclusions is that it is possible to generate biomaterial production processes with low costs and high economic returns. Low costs are more accessible to low-income families that might be needing biomeshes. We obtained a selling price of USD 0.16 ± 0.078.cm^−2^, USD 0.086 ± 0.043.cm^−2^, and USD 0.069 ± 0.035.cm^−2^ for production capacities of 5, 15 y 25 × 10^3^ BPU.mth^−1^, respectively. These values are below current published prices that range between USD 4 and 15.cm^−2^. The results of the present research will help slaughterhouses in Mexico to implement circular economy practices, reduce waste generation, and generate additional revenue.

## Figures and Tables

**Figure 1 polymers-15-02797-f001:**
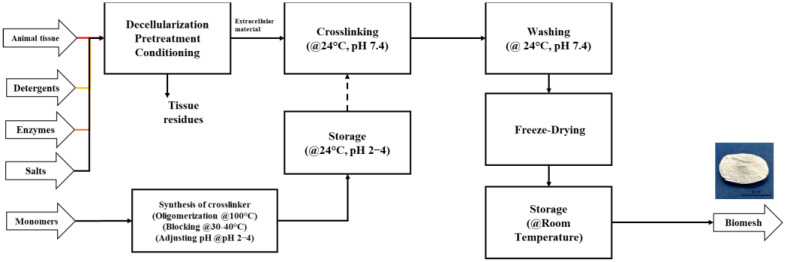
Process block diagram of BP biomesh production.

**Figure 2 polymers-15-02797-f002:**
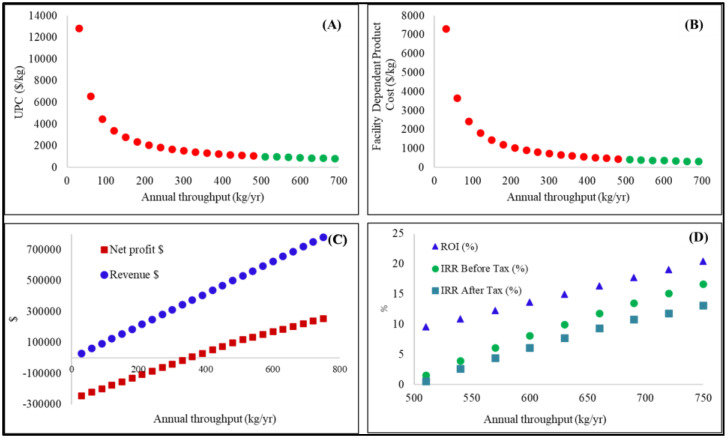
Profitability analysis for a process capacity of 5 × 10^3^ BPU.mth^−1^ : (**A**) Unit Production Cost UPC (USD/kg) vs. Annual throughput (kg/yr^−1^); (**B**) Facility dependent product cost (USD/kg^−1^) vs. annual throughput (kg.yr^−1^); (**C**) Net profit and revenue ($) vs. annual throughput (kg.yr^−1^); (**D**) ROI (%), IRR before tax (%) and IRR after tax (%) vs. annual throughput (kg.day^−1^). In Figure 2 (**A**,**B**) the red point and green point represent the unprofitability and profitability scenarios, respectively.

**Figure 3 polymers-15-02797-f003:**
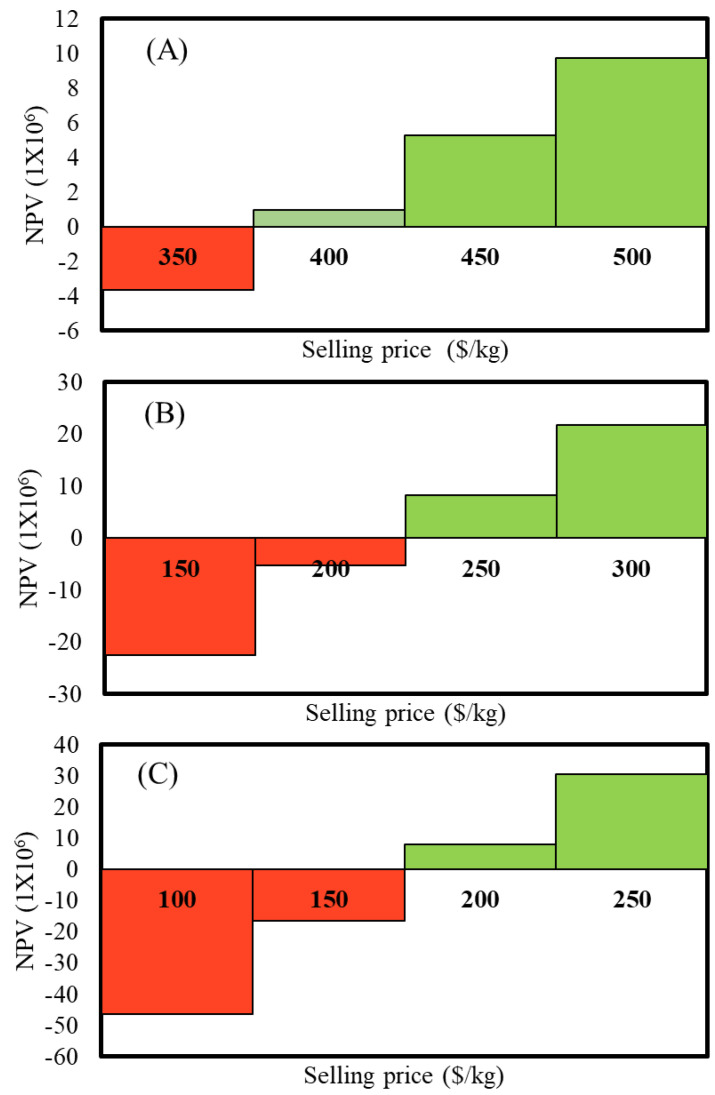
Net Present Value vs. biomesh selling price: (**A**) 5 × 10^3^ BPU/yr (2.5 kg of biomesh/batch); (**B**) 15 × 10^3^ BPU/yr (7.5 kg of biomesh/batch); (**C**) 25 × 10^3^ BPU/yr (12.5 kg of biomesh/batch).

**Figure 4 polymers-15-02797-f004:**
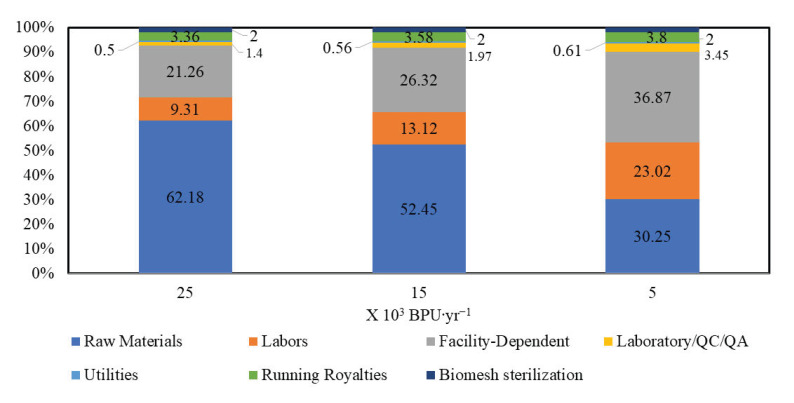
Distribution of costs in the biomesh production.

**Table 1 polymers-15-02797-t001:** Reactions involved in the CCLCS process production.

Equation	Oligomer Structure	Yield (%)	Equation #
Reactions involved in the synthesis of the crosslinker (oligomerization)			
2 HDI+1 PEG → 1 HDI−PEG−HDI	Trimer	70	(1)
3 HDI+2 PEG → 1 HDI−PEG−HDI−PEG−HDI	Pentamer	80	(2)
4 HDI+3 PEG → 1 HDI−PEG−HDI−PEG−HDI−PEG−HDI	Heptamer	99	(3)
Reactions used to block oligourethane			
$2\;NaHSO_{3}+1\;HDI\;\to$ $1\;Na^{+}SO_{3}^{-}-HDI-SO_{3}^{-}Na^{+}$		90	(4)
$2\;NaHSO_{3}+1\;Trimer\to$ $1Na^{+}SO_{3}^{-}-Trimer-SO_{3}^{-}Na^{+}$		90	(5)
$2\;NaHSO_{3}+1\;Pentamer\;\to$ $1\;Na^{+}SO_{3}^{-}-Pentamer-SO_{3}^{-}Na^{+}$		90	(6)
$2NaHSO_{3}+1\;Heptamer\to$ $1\;Na^{+}SO_{3}^{-}-Heptamer-SO_{3}^{-}Na^{+}$		90	(7)
Crosslinking reactions.			
$45\;Na^{+}SO_{3}^{-}-HDI-SO_{3}^{-}Na^{+}+1\;Collagen\;\to$ $1\;Collagen-{\left(HDI\right)}_{45}+90\;NaHSO_{3}$		90	(8)
$45\;Na^{+}SO_{3}^{-}-Trimer-SO_{3}^{-}Na^{+}+1\;Collagen\;\to$ $1\;Collagen-{\left(Trimer\right)}_{45}+90\;NaHSO_{3}$		90	(9)
$45\;Na^{+}SO_{3}^{-}-Pentamer-SO_{3}^{-}Na^{+}+1\;Collagen\;\to$ $1\;Collagen-{\left(Pentamer\right)}_{45}+90\;NaHSO_{3}$		90	(10)
$45\;Na^{+}SO_{3}^{-}-Heptamer-SO_{3}^{-}Na^{+}+1\;Collagen\;\to$ $1\;Collagen-{\left(Heptamer\right)}_{45}+90\;NaHSO_{3}$		90	(11)

**Table 2 polymers-15-02797-t002:** Investment summary for the manufacture of biomesh from BPU considering different scenarios.

	5000 BPU.mth^−1^	15,000 BPU.mth^−1^	25,000 BPU.mth^−1^	Units
Total capital investment	1,258,000	1,613,000	1,872,000	USD
Operating cost	612,560	1,074,320	1,515,280	USD/yr
Revenues	783,000	1,305,000	1,740,000	USD/yr
Cost basis annual rate	750.00	2,250	3,750	kg MP/yr
Net unit production cost	784.57	458.94	388.26	USD/kg MP
Unit production revenue	1,043.75	579.86	463.89	USD/kg MP
Unit production revenue	0.16 ± 0.078	0.086 ± 0.043	0.069 ± 0.035	USD/cm^2^ MP
Gross margin	24.79	20.85	16.30	%
Return on investment	20.38	21.31	19.90	%
Payback time	4.91	4.69	5.03	years
IRR (After taxes)	13.06	14.34	13.25	%
NPV (at 7.0% interest)	523,000	817,000	787,000	$

## Data Availability

Simulations can be obtained on request from the corresponding author C.E.MG.

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
