# Peer review of "Profitability of Chemically Cross-Linked Collagen Scaffold Production Using Bovine Pericardium: Revaluing Waste from the Meat Industry for Biomedical Applications"

_polymers, 2023, doi:10.3390/polym15132797_

Round 1
Reviewer 1 Report
This is a very exciting work! This manuscript presents an economic analysis of large-scale production of chemically cross-linked collagen scaffold (CCLCS) using waste generated by the meat industry. The author simulated three production capacities using SuperPro Designer® v 12.0, and demonstrated the feasibility of producing low-cost and highly profitable CCLCS through a continuous process using waste from the meat industry. Overall, this work showed the potential for stimulating a circular bioeconomy and indicate the economic viability of utilizing waste materials for the production of valuable bio-implants.
I strongly believe this manuscript will be insightful to both research and industry commnity. I highly recommend to accept this work.
Author Response
Thank you very much for the comment.
Reviewer 2 Report
Dear authors! In my opinion, the manuscript is more suitable for journals publishing works with a certain applied orientation, for example, Applied Sciences (MDPI). For publication in the Polymers, it is necessary to strengthen the "chemical" part. It is necessary to present chemical equations describing the formation of a crosslinking agent and the crosslinking process itself. Give the reaction data with the mechanism and justification. For the products obtained, it is necessary to present morphology or at least ordinary photos. If all the necessary recommendations are followed, this article can be accepted for consideration in this journal.
Author Response
Thank you for the comment. We have added a new Appendix A. In this Appendix we have placed the complete reaction that occur in the reactors during process. Please see Appendix A, Figure A1 and A2. In addition, please see page 4, line 128 to 133.
In addition, we have added a new Figure A3. In Figure A3 it is possible observe and ordinary photo and a micrography of the material. Please, see Figure A3.

Reviewer 3 Report
In the manuscript titled "Profitability of chemically cross-linked collagen scaffold production using bovine pericardium: revaluing waste from the meat industry for biomedical applications" the authors provide the detailed analysis for a novel large scale process, used for the production of crosslinked collagen scaffolds from natural materials such as wastes from meat industry. The work is well described and contextualized, provides convincing and detailed projections and discussions, and I find it very interesting as a concrete approach towards the employement of sustainable materials and processes at the industrial scale. I recommend this paper for the publication in Polymers in its current form.
Minor editing of English language required
Author Response
Thank you very much for the comment.
Round 2
Reviewer 2 Report
The authors have significantly strengthened the manuscript. There are no comments.